# Optimal Regret Minimization in Posted-Price Auctions with Strategic Buyers

**Mehryar Mohri**
Courant Institute and Google Research
251 Mercer Street
New York, NY 10012
mohri@cims.nyu.edu

**Andres Muñoz Medina**
Courant Institute
251 Mercer Street
New York, NY 10012
munoz@cims.nyu.edu

## Abstract

We study revenue optimization learning algorithms for posted-price auctions with strategic buyers. We analyze a very broad family of monotone regret minimization algorithms for this problem, which includes the previously best known algorithm, and show that no algorithm in that family admits a strategic regret more favorable than $\Omega(\sqrt{T})$. We then introduce a new algorithm that achieves a strategic regret differing from the lower bound only by a factor in $O(\log T)$, an exponential improvement upon the previous best algorithm. Our new algorithm admits a natural analysis and simpler proofs, and the ideas behind its design are general. We also report the results of empirical evaluations comparing our algorithm with the previous state of the art and show a consistent exponential improvement in several different scenarios.

## 1 Introduction

Auctions have long been an active area of research in Economics and Game Theory [Vickrey, 2012, Milgrom and Weber, 1982, Ostrovsky and Schwarz, 2011]. In the past decade, however, the advent of online advertisement has prompted a more algorithmic study of auctions, including the design of learning algorithms for revenue maximization for generalized second-price auctions or second-price auctions with reserve [Cesa-Bianchi et al., 2013, Mohri and Muñoz Medina, 2014, He et al., 2013].

These studies have been largely motivated by the widespread use of AdExchanges and the vast amount of historical data thereby collected – AdExchanges are advertisement selling platforms using second-price auctions with reserve price to allocate advertisement space. Thus far, the learning algorithms proposed for revenue maximization in these auctions critically rely on the assumption that the bids, that is, the outcomes of auctions, are drawn i.i.d. according to some unknown distribution. However, this assumption may not hold in practice. In particular, with the knowledge that a revenue optimization algorithm is being used, an advertiser could seek to mislead the publisher by under-bidding. In fact, consistent empirical evidence of strategic behavior by advertisers has been found by Edelman and Ostrovsky [2007]. This motivates the analysis presented in this paper of the interactions between sellers and *strategic buyers*, that is, buyers that may act non-truthfully with the goal of maximizing their surplus.

The scenario we consider is that of *posted-price auctions*, which, albeit simpler than other mechanisms, in fact matches a common situation in AdExchanges where many auctions admit a single bidder. In this setting, second-price auctions with reserve are equivalent to posted-price auctions: a seller sets a reserve price for a good and the buyer decides whether or not to accept it (that is to bid higher than the reserve price). In order to capture the buyer's strategic behavior, we will analyze an online scenario: at each time $t$, a price $p_t$ is offered by the seller and the buyer must decide to either accept it or leave it. This scenario can be modeled as a two-player repeated non-zero sum game with

incomplete information, where the seller's objective is to maximize his revenue, while the advertiser seeks to maximize her surplus as described in more detail in Section 2.

The literature on non-zero sum games is very rich [Nachbar, 1997, 2001, Morris, 1994], but much of the work in that area has focused on characterizing different types of equilibria, which is not directly relevant to the algorithmic questions arising here. Furthermore, the problem we consider admits a particular structure that can be exploited to design efficient revenue optimization algorithms.

From the seller's perspective, this game can also be viewed as a bandit problem [Kuleshov and Precup, 2010, Robbins, 1985] since only the revenue (or reward) for the prices offered is accessible to the seller. Kleinberg and Leighton [2003] precisely studied this continuous bandit setting under the assumption of an oblivious buyer, that is, one that does not exploit the seller's behavior (more precisely, the authors assume that at each round the seller interacts with a different buyer). The authors presented a tight regret bound of $\Theta(\log \log T)$ for the scenario of a buyer holding a fixed valuation and a regret bound of $O(T^{\frac{2}{3}})$ when facing an adversarial buyer by using an elegant reduction to a discrete bandit problem. However, as argued by Amin et al. [2013], when dealing with a *strategic buyer*, the usual definition of regret is no longer meaningful. Indeed, consider the following example: let the valuation of the buyer be given by $v \in [0, 1]$ and assume that an algorithm with sublinear regret such as Exp3 [Auer et al., 2002b] or UCB [Auer et al., 2002a] is used for $T$ rounds by the seller. A possible strategy for the buyer, knowing the seller's algorithm, would be to accept prices only if they are smaller than some small value $\epsilon$, certain that the seller would eventually learn to offer only prices less than $\epsilon$. If $\epsilon \ll v$, the buyer would considerably boost her surplus while, in theory, the seller would have not incurred a large regret since in hindsight, the best fixed strategy would have been to offer price $\epsilon$ for all rounds. This, however is clearly not optimal for the seller. The stronger notion of policy regret introduced by Arora et al. [2012] has been shown to be the appropriate one for the analysis of bandit problems with adaptive adversaries. However, for the example just described, a sublinear policy regret can be similarly achieved. Thus, this notion of regret is also not the pertinent one for the study of our scenario.

We will adopt instead the definition of *strategic-regret*, which was introduced by Amin et al. [2013] precisely for the study of this problem. This notion of regret also matches the concept of *learning loss* introduced by [Agrawal, 1995] when facing an oblivious adversary. Using this definition, Amin et al. [2013] presented both upper and lower bounds for the regret of a seller facing a strategic buyer and showed that the buyer's surplus must be discounted over time in order to be able to achieve sublinear regret (see Section 2). However, the gap between the upper and lower bounds they presented is in $O(\sqrt{T})$. In the following, we analyze a very broad family of monotone regret minimization algorithms for this problem (Section 3), which includes the algorithm of Amin et al. [2013], and show that no algorithm in that family admits a strategic regret more favorable than $\Omega(\sqrt{T})$. Next, we introduce a nearly-optimal algorithm that achieves a strategic regret differing from the lower bound at most by a factor in $O(\log T)$ (Section 4). This represents an exponential improvement upon the existing best algorithm for this setting. Our new algorithm admits a natural analysis and simpler proofs. A key idea behind its design is a method deterring the buyer from *lying*, that is rejecting prices below her valuation.

## 2 Setup

We consider the following game played by a buyer and a seller. A good, such as an advertisement space, is repeatedly offered for sale by the seller to the buyer over $T$ rounds. The buyer holds a private valuation $v \in [0, 1]$ for that good. At each round $t = 1, \ldots, T$, a price $p_t$ is offered by the seller and a decision $a_t \in \{0, 1\}$ is made by the buyer. $a_t$ takes value 1 when the buyer accepts to buy at that price, 0 otherwise. We will say that a buyer *lies* whenever $a_t = 0$ while $p_t < v$. At the beginning of the game, the algorithm $\mathcal{A}$ used by the seller to set prices is announced to the buyer. Thus, the buyer plays strategically against this algorithm. The knowledge of $\mathcal{A}$ is a standard assumption in mechanism design and also matches the practice in AdExchanges.

For any $\gamma \in (0, 1)$, define the discounted surplus of the buyer as follows:

$$\text{Sur}(\mathcal{A}, v) = \sum_{t=1}^{T} \gamma^{t-1} a_t(v - p_t). \tag{1}$$

The value of the *discount factor* $\gamma$ indicates the strength of the preference of the buyer for current surpluses versus future ones. The performance of a seller's algorithm is measured by the notion of *strategic-regret* [Amin et al., 2013] defined as follows:

$$\text{Reg}(\mathcal{A}, v) = Tv - \sum_{t=1}^{T} a_t p_t. \tag{2}$$

The buyer's objective is to maximize his discounted surplus, while the seller seeks to minimize his regret. Note that, in view of the discounting factor $\gamma$, the buyer is not fully adversarial. The problem consists of designing algorithms achieving sublinear strategic regret (that is a regret in $o(T)$).

The motivation behind the definition of strategic-regret is straightforward: a seller, with access to the buyer's valuation, can set a fixed price for the good $\epsilon$ close to this value. The buyer, having no control on the prices offered, has no option but to accept this price in order to optimize his utility. The revenue per round of the seller is therefore $v - \epsilon$. Since there is no scenario where higher revenue can be achieved, this is a natural setting to compare the performance of our algorithm.

To gain more intuition about the problem, let us examine some of the complications arising when dealing with a strategic buyer. Suppose the seller attempts to *learn* the buyer's valuation $v$ by performing a binary search. This would be a natural algorithm when facing a truthful buyer. However, in view of the buyer's knowledge of the algorithm, for $\gamma \gg 0$, it is in her best interest to lie on the initial rounds, thereby quickly, in fact exponentially, decreasing the price offered by the seller. The seller would then incur an $\Omega(T)$ regret. A binary search approach is therefore "too aggressive". Indeed, an untruthful buyer can manipulate the seller into offering prices less than $v/2$ by lying about her value even just once! This discussion suggests following a more conservative approach. In the next section, we discuss a natural family of conservative algorithms for this problem.

## 3 Monotone algorithms

The following conservative pricing strategy was introduced by Amin et al. [2013]. Let $p_1 = 1$ and $\beta < 1$. If price $p_t$ is rejected at round $t$, the lower price $p_{t+1} = \beta p_t$ is offered at the next round. If at any time price $p_t$ is accepted, then this price is offered for all the remaining rounds. We will denote this algorithm by `monotone`. The motivation behind its design is clear: for a suitable choice of $\beta$, the seller can slowly decrease the prices offered, thereby pressing the buyer to reject many prices (which is not convenient for her) before obtaining a favorable price. The authors present an $O(T_\gamma \sqrt{T})$ regret bound for this algorithm, with $T_\gamma = 1/(1-\gamma)$. A more careful analysis shows that this bound can be further tightened to $O(\sqrt{T_\gamma T} + \sqrt{T})$ when the discount factor $\gamma$ is known to the seller.

Despite its sublinear regret, the `monotone` algorithm remains sub-optimal for certain choices of $\gamma$. Indeed, consider a scenario with $\gamma \ll 1$. For this setting, the buyer would no longer have an incentive to lie, thus, an algorithm such as binary search would achieve logarithmic regret, while the regret achieved by the `monotone` algorithm is only guaranteed to be in $O(\sqrt{T})$.

One may argue that the `monotone` algorithm is too specific since it admits a single parameter $\beta$ and that perhaps a more complex algorithm with the same monotonic idea could achieve a more favorable regret. Let us therefore analyze a generic monotone algorithm $\mathcal{A}_m$ defined by Algorithm 1.

**Definition 1.** *For any buyer's valuation $v \in [0,1]$, define the* acceptance time $\kappa^* = \kappa^*(v)$ *as the first time a price offered by the seller using algorithm $\mathcal{A}_m$ is accepted.*

**Proposition 1.** *For any decreasing sequence of prices $(p_t)_{t=1}^{T}$, there exists a truthful buyer with valuation $v_0$ such that algorithm $\mathcal{A}_m$ suffers regret of at least*

$$Reg(\mathcal{A}_m, v_0) \geq \frac{1}{4} \sqrt{T - \sqrt{T}}.$$

*Proof.* By definition of the regret, we have $\text{Reg}(\mathcal{A}_m, v) = v\kappa^* + (T - \kappa^*)(v - p_{\kappa^*})$. We can consider two cases: $\kappa^*(v_0) > \sqrt{T}$ for some $v_0 \in [1/2, 1]$ and $\kappa^*(v) \leq \sqrt{T}$ for every $v \in [1/2, 1]$. In the former case, we have $\text{Reg}(\mathcal{A}_m, v_0) \geq v_0 \sqrt{T} \geq \frac{1}{2}\sqrt{T}$, which implies the statement of the proposition. Thus, we can assume the latter condition.

---
**Algorithm 1** Family of monotone algorithms.

> Let $p_1 = 1$ and $p_t \leq p_{t-1}$ for $t = 2, \ldots T$.
> $t \leftarrow 1$
> $p \leftarrow p_t$
> Offer price $p$
> **while** (Buyer rejects $p$) **and** $(t < T)$ **do**
>     $t \leftarrow t + 1$
>     $p \leftarrow p_t$
>     Offer price $p$
> **end while**
> **while** $(t < T)$ **do**
>     $t \leftarrow t + 1$
>     Offer price $p$
> **end while**

---
**Algorithm 2** Definition of $\mathcal{A}_r$.

> $\mathfrak{n} =$ the root of $\mathscr{T}(T)$
> **while** Offered prices less than $T$ **do**
>     Offer price $p_\mathfrak{n}$
>     **if** Accepted **then**
>         $\mathfrak{n} = r(\mathfrak{n})$
>     **else**
>         Offer price $p_n$ for $r$ rounds
>         $\mathfrak{n} = l(\mathfrak{n})$
>     **end if**
> **end while**

---

Let $v$ be uniformly distributed over $[\frac{1}{2}, 1]$. In view of Lemma 4 (see Appendix 8.1), we have

$$\mathbb{E}[v\kappa^*] + \mathbb{E}[(T - \kappa^*)(v - p_{\kappa^*})] \geq \frac{1}{2}\mathbb{E}[\kappa^*] + (T - \sqrt{T})\mathbb{E}[(v - p_{\kappa^*})] \geq \frac{1}{2}\mathbb{E}[\kappa^*] + \frac{T - \sqrt{T}}{32\mathbb{E}[\kappa^*]}.$$

The right-hand side is minimized for $\mathbb{E}[\kappa^*] = \frac{\sqrt{T - \sqrt{T}}}{4}$. Plugging in this value yields $\mathbb{E}[\text{Reg}(\mathcal{A}_m, v)] \geq \frac{\sqrt{T - \sqrt{T}}}{4}$, which implies the existence of $v_0$ with $\text{Reg}(\mathcal{A}_m, v_0) \geq \frac{\sqrt{T - \sqrt{T}}}{4}$. $\qquad\square$

We have thus shown that any monotone algorithm $\mathcal{A}_m$ suffers a regret of at least $\Omega(\sqrt{T})$, even when facing a truthful buyer. A tighter lower bound can be given under a mild condition on the prices offered.

**Definition 2.** *A sequence $(p_t)_{t=1}^T$ is said to be* convex *if it verifies $p_t - p_{t+1} \geq p_{t+1} - p_{t+2}$ for $t = 1, \ldots, T - 2$.*

An instance of a convex sequence is given by the prices offered by the `monotone` algorithm. A seller offering prices forming a decreasing convex sequence seeks to control the number of lies of the buyer by slowly reducing prices. The following proposition gives a lower bound on the regret of any algorithm in this family.

**Proposition 2.** *Let $(p_t)_{t=1}^T$ be a decreasing convex sequence of prices. There exists a valuation $v_0$ for the buyer such that the regret of the monotone algorithm defined by these prices is $\Omega(\sqrt{TC_\gamma} + \sqrt{T})$, where $C_\gamma = \frac{\gamma}{2(1-\gamma)}$.*

The full proof of this proposition is given in Appendix 8.1. The proposition shows that when the discount factor $\gamma$ is known, the `monotone` algorithm is in fact asymptotically optimal in its class.

The results just presented suggest that the dependency on $T$ cannot be improved by any monotone algorithm. In some sense, this family of algorithms is "too conservative". Thus, to achieve a more favorable regret guarantee, an entirely different algorithmic idea must be introduced. In the next section, we describe a new algorithm that achieves a substantially more advantageous strategic regret by combining the fast convergence properties of a binary search-type algorithm (in a truthful setting) with a method penalizing untruthful behaviors of the buyer.

## 4    A nearly optimal algorithm

Let $\mathcal{A}$ be an algorithm for revenue optimization used against a truthful buyer. Denote by $\mathscr{T}(T)$ the tree associated to $\mathcal{A}$ after $T$ rounds. That is, $\mathscr{T}(T)$ is a full tree of height $T$ with nodes $\mathfrak{n} \in \mathscr{T}(T)$ labeled with the prices $p_\mathfrak{n}$ offered by $\mathcal{A}$. The right and left children of $\mathfrak{n}$ are denoted by $r(\mathfrak{n})$ and $l(\mathfrak{n})$ respectively. The price offered when $p_\mathfrak{n}$ is accepted by the buyer is the label of $r(\mathfrak{n})$ while the price offered by $\mathcal{A}$ if $p_\mathfrak{n}$ is rejected is the label of $l(\mathfrak{n})$. Finally, we will denote the left and right subtrees rooted at node $\mathfrak{n}$ by $\mathscr{L}(\mathfrak{n})$ and $\mathscr{R}(\mathfrak{n})$ respectively. Figure 1 depicts the tree generated by an algorithm proposed by Kleinberg and Leighton [2003], which we will describe later.

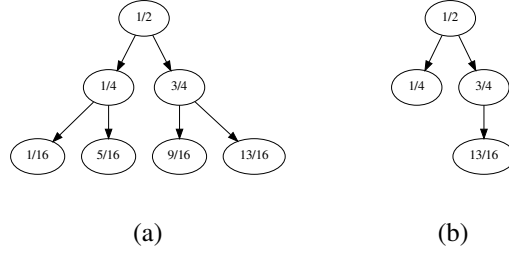

(a)                              (b)

Figure 1: (a) Tree $\mathscr{T}(3)$ associated to the algorithm proposed in [Kleinberg and Leighton, 2003]. (b) Modified tree $\mathscr{T}'(3)$ with $r = 2$.

Since the buyer holds a fixed valuation, we will consider algorithms that increase prices only after a price is accepted and decrease it only after a rejection. This is formalized in the following definition.

**Definition 3.** *An algorithm $\mathcal{A}$ is said to be* consistent *if* $\max_{\mathfrak{n}' \in \mathscr{L}(\mathfrak{n})} p_{\mathfrak{n}'} \leq p_{\mathfrak{n}} \leq \min_{\mathfrak{n}' \in \mathscr{R}(\mathfrak{n})} p_{\mathfrak{n}'}$ *for any node $\mathfrak{n} \in \mathscr{T}(T)$.*

For any consistent algorithm $\mathcal{A}$, we define a modified algorithm $\mathcal{A}_r$, parametrized by an integer $r \geq 1$, designed to face strategic buyers. Algorithm $\mathcal{A}_r$ offers the same prices as $\mathcal{A}$, but it is defined with the following modification: when a price is rejected by the buyer, the seller offers the same price for $r$ rounds. The pseudocode of $\mathcal{A}_r$ is given in Algorithm 2. The motivation behind the modified algorithm is given by the following simple observation: a strategic buyer will lie only if she is certain that rejecting a price will boost her surplus in the future. By forcing the buyer to reject a price for several rounds, the seller ensures that the future discounted surplus will be negligible, thereby coercing the buyer to be truthful.

We proceed to formally analyze algorithm $\mathcal{A}_r$. In particular, we will quantify the effect of the parameter $r$ on the choice of the buyer's strategy. To do so, a measure of the spread of the prices offered by $\mathcal{A}_r$ is needed.

**Definition 4.** *For any node $\mathfrak{n} \in \mathscr{T}(T)$ define the right increment of $\mathfrak{n}$ as $\delta_{\mathfrak{n}}^r := p_{r(\mathfrak{n})} - p_{\mathfrak{n}}$. Similarly, define its left increment to be $\delta_{\mathfrak{n}}^l := \max_{\mathfrak{n}' \in \mathscr{L}(\mathfrak{n})} p_{\mathfrak{n}} - p_{\mathfrak{n}'}$.*

The prices offered by $\mathcal{A}_r$ define a path in $\mathscr{T}(T)$. For each node in this path, we can define time $t(\mathfrak{n})$ to be the number of rounds needed for this node to be reached by $\mathcal{A}_r$. Note that, since $r$ may be greater than 1, the path chosen by $\mathcal{A}_r$ might not necessarily reach the leaves of $\mathscr{T}(T)$. Finally, let $\mathcal{S} \colon \mathfrak{n} \mapsto \mathcal{S}(\mathfrak{n})$ be the function representing the surplus obtained by the buyer when playing an optimal strategy against $\mathcal{A}_r$ after node $\mathfrak{n}$ is reached.

**Lemma 1.** *The function $\mathcal{S}$ satisfies the following recursive relation:*

$$\mathcal{S}(\mathfrak{n}) = \max(\gamma^{t(\mathfrak{n})-1}(v - p_{\mathfrak{n}}) + \mathcal{S}(r(\mathfrak{n})), \mathcal{S}(l(\mathfrak{n}))). \tag{3}$$

*Proof.* Define a weighted tree $\mathscr{T}'(T) \subset \mathscr{T}(T)$ of nodes reachable by algorithm $\mathcal{A}_r$. We assign weights to the edges in the following way: if an edge on $\mathscr{T}'(T)$ is of the form $(\mathfrak{n}, r(\mathfrak{n}))$, its weight is set to be $\gamma^{t(\mathfrak{n})-1}(v - p_{\mathfrak{n}})$, otherwise, it is set to 0. It is easy to see that the function $\mathcal{S}$ evaluates the weight of the longest path from node $\mathfrak{n}$ to the leafs of $\mathscr{T}'(T)$. It thus follows from elementary graph algorithms that equation (3) holds. $\square$

The previous lemma immediately gives us necessary conditions for a buyer to reject a price.

**Proposition 3.** *For any reachable node $\mathfrak{n}$, if price $p_{\mathfrak{n}}$ is rejected by the buyer, then the following inequality holds:*

$$v - p_{\mathfrak{n}} < \frac{\gamma^r}{(1-\gamma)(1-\gamma^r)}(\delta_{\mathfrak{n}}^l + \gamma \delta_{\mathfrak{n}}^r).$$

*Proof.* A direct implication of Lemma 1 is that price $p_{\mathfrak{n}}$ will be rejected by the buyer if and only if

$$\gamma^{t(\mathfrak{n})-1}(v - p_{\mathfrak{n}}) + \mathcal{S}(r(\mathfrak{n})) < \mathcal{S}(l(\mathfrak{n})). \tag{4}$$

However, by definition, the buyer's surplus obtained by following any path in $\mathscr{R}(\mathfrak{n})$ is bounded above by $\mathcal{S}(r(\mathfrak{n}))$. In particular, this is true for the path which rejects $p_{r(\mathfrak{n})}$ and accepts every price afterwards. The surplus of this path is given by $\sum_{t=t(\mathfrak{n})+r+1}^{T} \gamma^{t-1}(v - \widehat{p}_t)$ where $(\widehat{p}_t)_{t=t(\mathfrak{n})+r+1}^{T}$ are the prices the seller would offer if price $p_{r(\mathfrak{n})}$ were rejected. Furthermore, since algorithm $\mathcal{A}_r$ is consistent, we must have $\widehat{p}_t \leq p_{r(\mathfrak{n})} = p_{\mathfrak{n}} + \delta_{\mathfrak{n}}^r$. Therefore, $\mathcal{S}(r(\mathfrak{n}))$ can be bounded as follows:

$$\mathcal{S}(r(\mathfrak{n})) \geq \sum_{t=t(\mathfrak{n})+r+1}^{T} \gamma^{t-1}(v - p_{\mathfrak{n}} - \delta_{\mathfrak{n}}^r) = \frac{\gamma^{t(\mathfrak{n})+r} - \gamma^T}{1 - \gamma}(v - p_{\mathfrak{n}} - \delta_{\mathfrak{n}}^r). \qquad (5)$$

We proceed to upper bound $\mathcal{S}(l(\mathfrak{n}))$. Since $p_{\mathfrak{n}} - p'_n \leq \delta_{\mathfrak{n}}^l$ for all $\mathfrak{n}' \in \mathscr{L}(\mathfrak{n})$, $v - p_{\mathfrak{n}'} \leq v - p_{\mathfrak{n}} + \delta_{\mathfrak{n}}^l$ and

$$\mathcal{S}(l(\mathfrak{n})) \leq \sum_{t=t_{\mathfrak{n}}+r}^{T} \gamma^{t-1}(v - p_{\mathfrak{n}} + \delta_{\mathfrak{n}}^l) = \frac{\gamma^{t(\mathfrak{n})+r-1} - \gamma^T}{1 - \gamma}(v - p_{\mathfrak{n}} + \delta_{\mathfrak{n}}^l). \qquad (6)$$

Combining inequalities (4), (5) and (6) we conclude that

$$\gamma^{t(\mathfrak{n})-1}(v - p_{\mathfrak{n}}) + \frac{\gamma^{t(\mathfrak{n})+r} - \gamma^T}{1 - \gamma}(v - p_{\mathfrak{n}} - \delta_{\mathfrak{n}}^r) \leq \frac{\gamma^{t(\mathfrak{n})+r-1} - \gamma^T}{1 - \gamma}(v - p_{\mathfrak{n}} + \delta_{\mathfrak{n}}^l)$$

$$\Rightarrow \quad (v - p_{\mathfrak{n}})\left(1 + \frac{\gamma^{r+1} - \gamma^r}{1 - \gamma}\right) \leq \frac{\gamma^r \delta_{\mathfrak{n}}^l + \gamma^{r+1}\delta_{\mathfrak{n}}^r - \gamma^{T-t(\mathfrak{n})+1}(\delta_{\mathfrak{n}}^r + \delta_{\mathfrak{n}}^l)}{1 - \gamma}$$

$$\Rightarrow \quad (v - p_n)(1 - \gamma^r) \leq \frac{\gamma^r(\delta_{\mathfrak{n}}^l + \gamma \delta_{\mathfrak{n}}^r)}{1 - \gamma}.$$

Rearranging the terms in the above inequality yields the desired result. $\qquad \square$

Let us consider the following instantiation of algorithm $\mathcal{A}$ introduced in [Kleinberg and Leighton, 2003]. The algorithm keeps track of a *feasible interval* $[a, b]$ initialized to $[0, 1]$ and an increment parameter $\epsilon$ initialized to $1/2$. The algorithm works in phases. Within each phase, it offers prices $a + \epsilon, a + 2\epsilon, \ldots$ until a price is rejected. If price $a + k\epsilon$ is rejected, then a new phase starts with the feasible interval set to $[a + (k-1)\epsilon, a + k\epsilon]$ and the increment parameter set to $\epsilon^2$. This process continues until $b - a < 1/T$ at which point the last phase starts and price $a$ is offered for the remaining rounds. It is not hard to see that the number of phases needed by the algorithm is less than $\lceil \log_2 \log_2 T \rceil + 1$. A more surprising fact is that this algorithm has been shown to achieve regret $O(\log \log T)$ when the seller faces a truthful buyer. We will show that the modification $\mathcal{A}_r$ of this algorithm admits a particularly favorable regret bound. We will call this algorithm $\mathsf{PFS}_r$ (penalized fast search algorithm).

**Proposition 4.** *For any value of $v \in [0, 1]$ and any $\gamma \in (0, 1)$, the regret of algorithm $\mathsf{PFS}_r$ admits the following upper bound:*

$$Reg(\mathsf{PFS}_r, v) \leq (vr + 1)(\lceil \log_2 \log_2 T \rceil + 1) + \frac{(1 + \gamma)\gamma^r T}{2(1 - \gamma)(1 - \gamma^r)}. \qquad (7)$$

Note that for $r = 1$ and $\gamma \to 0$ the upper bound coincides with that of [Kleinberg and Leighton, 2003].

*Proof.* Algorithm $\mathsf{PFS}_r$ can accumulate regret in two ways: the price offered $p_{\mathfrak{n}}$ is rejected, in which case the regret is $v$, or the price is accepted and its regret is $v - p_{\mathfrak{n}}$.

Let $K = \lceil \log_2 \log_2 T \rceil + 1$ be the number of phases run by algorithm $\mathsf{PFS}_r$. Since at most $K$ different prices are rejected by the buyer (one rejection per phase) and each price must be rejected for $r$ rounds, the cumulative regret of all rejections is upper bounded by $vKr$.

The second type of regret can also be bounded straightforwardly. For any phase $i$, let $\epsilon_i$ and $[a_i, b_i]$ denote the corresponding search parameter and feasible interval respectively. If $v \in [a_i, b_i]$, the regret accrued in the case where the buyer accepts a price in this interval is bounded by $b_i - a_i = \sqrt{\epsilon_i}$. If, on the other hand $v \geq b_i$, then it readily follows that $v - p_{\mathfrak{n}} < v - b_i + \sqrt{\epsilon_i}$ for all prices $p_{\mathfrak{n}}$ offered in phase $i$. Therefore, the regret obtained in acceptance rounds is bounded by

$$\sum_{i=1}^{K} N_i\left((v - b_i)\mathbb{1}_{v > b_i} + \sqrt{\epsilon_i}\right) \leq \sum_{i=1}^{K}(v - b_i)\mathbb{1}_{v > b_i}N_i + K,$$

where $N_i \leq \frac{1}{\sqrt{\epsilon_i}}$ denotes the number of prices offered during the $i$-th round.

Finally, notice that, in view of the algorithm's definition, every $b_i$ corresponds to a rejected price. Thus, by Proposition 3, there exist nodes $\mathfrak{n}_i$ (not necessarily distinct) such that $p_{\mathfrak{n}_i} = b_i$ and

$$v - b_i = v - p_{\mathfrak{n}_i} \leq \frac{\gamma^r}{(1-\gamma)(1-\gamma^r)} (\delta^l_{\mathfrak{n}_i} + \gamma \delta^r_{n_i}).$$

It is immediate that $\delta^r_{\mathfrak{n}} \leq 1/2$ and $\delta^l_{\mathfrak{n}} \leq 1/2$ for any node $\mathfrak{n}$, thus, we can write

$$\sum_{i=1}^{K}(v - b_i)\mathbb{1}_{v>b_i} N_i \leq \frac{\gamma^r(1+\gamma)}{2(1-\gamma)(1-\gamma^r)} \sum_{i=1}^{K} N_i \leq \frac{\gamma^r(1+\gamma)}{2(1-\gamma)(1-\gamma^r)} T.$$

The last inequality holds since at most $T$ prices are offered by our algorithm. Combining the bounds for both regret types yields the result. $\square$

When an upper bound on the discount factor $\gamma$ is known to the seller, he can leverage this information and optimize upper bound (7) with respect to the parameter $r$.

**Theorem 1.** *Let $1/2 < \gamma < \gamma_0 < 1$ and $r^* = \left\lceil \operatorname{argmin}_{r \geq 1} r + \frac{\gamma_0^r T}{(1-\gamma_0)(1-\gamma_0^r)} \right\rceil$. For any $v \in [0,1]$, if $T > 4$, the regret of $\mathsf{PFS}_{r*}$ satisfies*

$$Reg(\mathsf{PFS}_{r*}, v) \leq (2v\gamma_0 T_{\gamma_0} \log cT + 1 + v)(\log_2 \log_2 T + 1) + 4T_{\gamma_0},$$

*where $c = 4 \log 2$.*

The proof of this theorem is fairly technical and is deferred to the Appendix. The theorem helps us define conditions under which logarithmic regret can be achieved. Indeed, if $\gamma_0 = e^{-1/\log T} = O(1 - \frac{1}{\log T})$, using the inequality $e^{-x} \leq 1 - x + x^2/2$ valid for all $x > 0$ we obtain

$$\frac{1}{1-\gamma_0} \leq \frac{\log^2 T}{2\log T - 1} \leq \log T.$$

It then follows from Theorem 1 that

$$\mathrm{Reg}(\mathsf{PFS}_{r*}, v) \leq (2v \log T \log cT + 1 + v)(\log_2 \log_2 T + 1) + 4\log T.$$

Let us compare the regret bound given by Theorem 1 with the one given by Amin et al. [2013]. The above discussion shows that for certain values of $\gamma$, an exponentially better regret can be achieved by our algorithm. It can be argued that the knowledge of an upper bound on $\gamma$ is required, whereas this is not needed for the monotone algorithm. However, if $\gamma > 1 - 1/\sqrt{T}$, the regret bound on monotone is super-linear, and therefore uninformative. Thus, in order to properly compare both algorithms, we may assume that $\gamma < 1 - 1/\sqrt{T}$ in which case, by Theorem 1, the regret of our algorithm is $O(\sqrt{T} \log T)$ whereas only linear regret can be guaranteed by the monotone algorithm. Even under the more favorable bound of $O(\sqrt{T_\gamma T} + \sqrt{T})$, for any $\alpha < 1$ and $\gamma < 1 - 1/T^\alpha$, the monotone algorithm will achieve regret $O(T^{\frac{\alpha+1}{2}})$ while a strictly better regret $O(T^\alpha \log T \log \log T)$ is attained by ours.

## 5 Lower bound

The following lower bounds have been derived in previous work.

**Theorem 2** ([Amin et al., 2013])**.** *Let $\gamma > 0$ be fixed. For any algorithm $\mathcal{A}$, there exists a valuation $v$ for the buyer such that $Reg(\mathcal{A}, v) \geq \frac{1}{12}T_\gamma$.*

This theorem is in fact given for the stochastic setting where the buyer's valuation is a random variable taken from some fixed distribution $\mathcal{D}$. However, the proof of the theorem selects $\mathcal{D}$ to be a point mass, therefore reducing the scenario to a fixed priced setting.

**Theorem 3** ( [Kleinberg and Leighton, 2003])**.** *Given any algorithm $\mathcal{A}$ to be played against a truthful buyer, there exists a value $v \in [0,1]$ such that $Reg(\mathcal{A}, v) \geq C \log \log T$ for some universal constant $C$.*

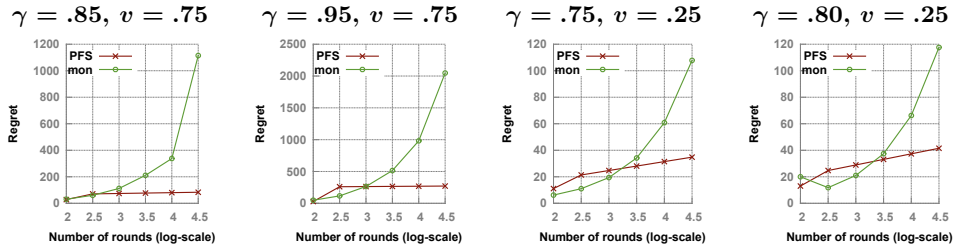

Figure 2: Comparison of the `monotone` algorithm and $\mathsf{PFS}_r$ for different choices of $\gamma$ and $v$. The regret of each algorithm is plotted as a function of the number rounds when $\gamma$ is not known to the algorithms (first two figures) and when its value is made accessible to the algorithms (last two figures).

Combining these results leads immediately to the following.

**Corollary 1.** *Given any algorithm $\mathcal{A}$, there exists a buyer's valuation $v \in [0,1]$ such that $Reg(\mathcal{A}, v) \geq \max\left(\frac{1}{12}T_\gamma, C\log\log T\right)$, for a universal constant $C$.*

We now compare the upper bounds given in the previous section with the bound of Corollary 1. For $\gamma > 1/2$, we have $\mathrm{Reg}(\mathsf{PFS}_r, v) = O(T_\gamma \log T \log\log T)$. On the other hand, for $\gamma \leq 1/2$, we may choose $r = 1$, in which case, by Proposition 4, $\mathrm{Reg}(\mathsf{PFS}_r, v) = O(\log\log T)$. Thus, the upper and lower bounds match up to an $O(\log T)$ factor.

## 6 Empirical results

In this section, we present the result of simulations comparing the `monotone` algorithm and our algorithm $\mathsf{PFS}_r$. The experiments were carried out as follows: given a buyer's valuation $v$, a discrete set of false valuations $\widehat{v}$ were selected out of the set $\{.03, .06, \ldots, v\}$. Both algorithms were run against a buyer making the seller believe her valuation is $\widehat{v}$ instead of $v$. The value of $\widehat{v}$ achieving the best utility for the buyer was chosen and the regret for both algorithms is reported in Figure 2.

We considered two sets of experiments. First, the value of parameter $\gamma$ was left unknown to both algorithms and the value of $r$ was set to $\log(T)$. This choice is motivated by the discussion following Theorem 1 since, for large values of $T$, we can expect to achieve logarithmic regret. The first two plots (from left to right) in Figure 2 depict these results. The apparent stationarity in the regret of $\mathsf{PFS}_r$ is just a consequence of the scale of the plots as the regret is in fact growing as $\log(T)$. For the second set of experiments, we allowed access to the parameter $\gamma$ to both algorithms. The value of $r$ was chosen optimally based on the results of Theorem 1 and the parameter $\beta$ of `monotone` was set to $1 - 1/\sqrt{TT_\gamma}$ to ensure regret in $O(\sqrt{TT_\gamma} + \sqrt{T})$. It is worth noting that even though our algorithm was designed under the assumption of some knowledge about the value of $\gamma$, the experimental results show that an exponentially better performance over the `monotone` algorithm is still attainable and in fact the performances of the optimized and unoptimized versions of our algorithm are comparable. A more comprehensive series of experiments is presented in Appendix 9.

## 7 Conclusion

We presented a detailed analysis of revenue optimization algorithms against strategic buyers. In doing so, we reduced the gap between upper and lower bounds on strategic regret to a logarithmic factor. Furthermore, the algorithm we presented is simple to analyze and reduces to the truthful scenario in the limit of $\gamma \to 0$, an important property that previous algorithms did not admit. We believe that our analysis helps gain a deeper understanding of this problem and that it can serve as a tool for studying more complex scenarios such as that of strategic behavior in repeated second-price auctions, VCG auctions and general market strategies.

## Acknowledgments

We thank Kareem Amin, Afshin Rostamizadeh and Umar Syed for several discussions about the topic of this paper. This work was partly funded by the NSF award IIS-1117591.

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
