[Supplementary Material]

# 8 Appendix

**Lemma 2.** *The function $g : \gamma \mapsto \frac{\log \frac{1}{\gamma}}{1-\gamma}$ is decreasing over the interval $(0,1)$.*

*Proof.* This can be straightforwardly established:

$$g'(\gamma) = \frac{-\frac{1-\gamma}{\gamma} + \log \frac{1}{\gamma}}{(1-\gamma)^2} = \frac{\gamma \log \left(1 - \left[1 - \frac{1}{\gamma}\right]\right) - (1-\gamma)}{\gamma(1-\gamma)^2} < \frac{(1-\gamma) - (1-\gamma)}{\gamma(1-\gamma)^2} = 0,$$

using the inequality $\log(1-x) < -x$ valid for all $x < 0$. $\qquad\square$

**Lemma 3.** *Let $a \geq 0$ and let $g \colon D \subset \mathbb{R} \to [a, \infty)$ be a decreasing and differentiable function. Then, the function $F \colon \mathbb{R} \to \mathbb{R}$ defined by*

$$F(\gamma) = g(\gamma) - \sqrt{g(\gamma)^2 - b}$$

*is increasing for all values of $b \in [0, a]$.*

*Proof.* We will show that $F'(\gamma) \geq 0$ for all $\gamma \in D$. Since $F' = g'[1 - g(g^2 - b)^{-1/2}]$ and $g' \leq 0$ by hypothesis, the previous statement is equivalent to showing that $\sqrt{g^2 - b} \leq g$ which is trivially verified since $b \geq 0$. $\qquad\square$

**Theorem 1.** *Let $1/2 < \gamma < \gamma_0 < 1$ and $r^* = \left\lceil \operatorname{argmin}_{r \geq 1} r + \frac{\gamma_0^r T}{(1-\gamma_0)(1-\gamma_0^r)} \right\rceil$. For any $v \in [0,1]$, if $T > 4$, the regret of $\mathsf{PFS}_{r^*}$ satisfies*

$$Reg(\mathsf{PFS}_{r^*}, v) \leq (2v\gamma_0 T_{\gamma_0} \log cT + 1 + v)(\log_2 \log_2 T + 1) + 4T_{\gamma_0},$$

*where $c = 4\log 2$.*

*Proof.* It is not hard to verify that the function $r \mapsto r + \frac{\gamma_0^r T}{(1-\gamma_0)(1-\gamma_0^r)}$ is convex and approaches infinity as $r \to \infty$. Thus, it admits a minimizer $\bar{r}^*$ whose explicit expression can be found by solving the following equation

$$0 = \frac{d}{dr} \left( r + \frac{\gamma_0^r T}{(1-\gamma_0)(1-\gamma_0^r)} \right) = 1 + \frac{\gamma_0^r T \log \gamma_0}{(1-\gamma_0)(1-\gamma_0^r)^2}.$$

Solving the corresponding second-degree equation yields

$$\gamma_0^{\bar{r}^*} = \frac{2 + \frac{T \log(1/\gamma_0)}{1-\gamma_0} - \sqrt{\left(2 + \frac{T \log(1/\gamma_0)}{1-\gamma_0}\right)^2 - 4}}{2} =: F(\gamma_0).$$

By Lemmas 2 and 3, the function $F$ thereby defined is increasing. Therefore, $\gamma_0^{\bar{r}^*} \leq \lim_{\gamma_0 \to 1} F(\gamma_0)$ and

$$\gamma_0^{\bar{r}^*} \leq \frac{2 + T - \sqrt{(2+T)^2 - 4}}{2} = \frac{4}{2(2 + T + \sqrt{(2+T)^2 - 4})} \leq \frac{2}{T}. \qquad (8)$$

By the same argument, we must have $\gamma_0^{\bar{r}^*} \geq F(1/2)$, that is

$$\begin{aligned}
\gamma_0^{\bar{r}^*} \geq F(1/2) &= \frac{2 + 2T \log 2 - \sqrt{(2 + 2T \log 2)^2 - 4}}{2} \\
&= \frac{4}{2(2 + 2T \log 2 + \sqrt{(2 + 2T \log 2)^2 - 4})} \\
&\geq \frac{2}{4 + 4T \log 2} \geq \frac{1}{4T \log 2}.
\end{aligned}$$

Thus,

$$r^* = \lceil \bar{r}^* \rceil \leq \frac{\log(1/F(1/2))}{\log(1/\gamma_0)} + 1 \leq \frac{\log(4T \log 2)}{\log 1/\gamma_0} + 1. \qquad (9)$$

Combining inequalities (8) and (9) with (7) gives

$$\text{Reg}(\text{PFS}_{r^*}, v) \leq \left( v \frac{\log(4T \log 2)}{\log 1/\gamma_0} + 1 + v \right) (\lceil \log_2 \log_2 T \rceil + 1) + \frac{(1 + \gamma_0)T}{(1 - \gamma_0)(T - 2)}$$
$$\leq (2v\gamma_0 T_{\gamma_0} \log(cT) + 1 + v)(\lceil \log_2 \log_2 T \rceil + 1) + 4T_{\gamma_0},$$

using the inequality $\log(\frac{1}{\gamma}) \geq \frac{1-\gamma}{2\gamma}$ valid for all $\gamma \in (1/2, 1)$. $\quad\square$

## 8.1 Lower bound for monotone algorithms

**Lemma 4.** *Let $(p_t)_{t=1}^T$ be a decreasing sequence of prices. Assume that the seller faces a truthful buyer. Then, if $v$ is sampled uniformly at random in the interval $[\frac{1}{2}, 1]$, the following inequality holds:*

$$\mathbb{E}[\kappa^*] \geq \frac{1}{32\mathbb{E}[v - p_{\kappa^*}]}.$$

*Proof.* Since the buyer is truthful, $\kappa^*(v) = \kappa$ if and only if $v \in [p_\kappa, p_{\kappa-1}]$. Thus, we can write

$$\mathbb{E}[v - p_{\kappa^*}] = \sum_{\kappa=2}^{\kappa_{\max}} \mathbb{E}\left[ \mathbb{1}_{v \in [p_\kappa, p_{\kappa-1}]}(v - p_\kappa) \right] = \sum_{\kappa=2}^{\kappa_{\max}} \int_{p_\kappa}^{p_{\kappa-1}} (v - p_\kappa)\, dv = \sum_{\kappa=2}^{\kappa_{\max}} \frac{(p_{\kappa-1} - p_\kappa)^2}{2},$$

where $\kappa_{\max} = \kappa^*(\frac{1}{2})$. Thus, by the Cauchy-Schwarz inequality, we can write

$$\mathbb{E}\left[ \sum_{\kappa=2}^{\kappa^*} p_{\kappa-1} - p_\kappa \right] \leq \mathbb{E}\left[ \sqrt{\kappa^* \sum_{\kappa=2}^{\kappa^*} (p_{\kappa-1} - p_\kappa)^2} \right]$$
$$\leq \mathbb{E}\left[ \sqrt{\kappa^* \sum_{\kappa=2}^{\kappa_{\max}} (p_{\kappa-1} - p_\kappa)^2} \right]$$
$$= \mathbb{E}\left[ \sqrt{2\kappa^* \mathbb{E}[v - p_{\kappa^*}]} \right]$$
$$\leq \sqrt{\mathbb{E}[\kappa^*]}\sqrt{2\mathbb{E}[v - p_\kappa^*]},$$

where the last step holds by Jensen's inequality. In view of that, since $v > p_{\kappa^*}$, it follows that:

$$\frac{3}{4} = \mathbb{E}[v] \geq \mathbb{E}[p_{\kappa^*}] = \mathbb{E}\left[ \sum_{\kappa=2}^{\kappa^*} p_\kappa - p_{\kappa-1} \right] + p_1 \geq -\sqrt{\mathbb{E}[\kappa^*]}\sqrt{2\mathbb{E}[v - p_{\kappa^*}]} + 1.$$

Solving for $\mathbb{E}[\kappa^*]$ concludes the proof. $\quad\square$

The following lemma characterizes the value of $\kappa^*$ when facing a strategic buyer.

**Lemma 5.** *For any $v \in [0, 1]$, $\kappa^*$ satisfies $v - p_{\kappa^*} \geq C_\gamma^{\kappa^*}(p_{\kappa^*} - p_{\kappa^*+1})$ with $C_\gamma^{\kappa^*} = \frac{\gamma - \gamma^{T-\kappa^*+1}}{1-\gamma}$. Furthermore, when $\kappa^* \leq 1 + \sqrt{T_\gamma T}$ and $T \geq T_\gamma + \frac{2\log(2/\gamma)}{\log(1/\gamma)}$, $C_\gamma^{\kappa^*}$ can be replaced by the universal constant $C_\gamma = \frac{\gamma}{2(1-\gamma)}$.*

*Proof.* Since an optimal strategy is played by the buyer, the surplus obtained by accepting a price at time $\kappa^*$ must be greater than the corresponding surplus obtained when accepting the first price at time $\kappa^* + 1$. It thus follows that:

$$\sum_{t=\kappa^*}^T \gamma^{t-1}(v - p_{\kappa^*}) \geq \sum_{t=\kappa^*+1}^T \gamma^{t-1}(v - p_{\kappa^*+1})$$
$$\Rightarrow \gamma^{\kappa^*-1}(v - p_{\kappa^*}) \geq \sum_{t=\kappa^*+1}^T \gamma^{t-1}(p_{\kappa^*} - p_{\kappa^*+1}) = \frac{\gamma^{\kappa^*} - \gamma^T}{1 - \gamma}(p_{\kappa^*} - p_{\kappa^*+1}).$$

Dividing both sides of the inequality by $\gamma^{\kappa^*-1}$ yields the first statement of the lemma. Let us verify the second statement. A straightforward calculation shows that the conditions on $T$ imply $T - \sqrt{TT_\gamma} \geq \frac{\log(2/\gamma)}{\log(1/\gamma)}$, therefore

$$C_\gamma^{\kappa^*} \geq \frac{\gamma - \gamma^{T-\sqrt{T_\gamma T}}}{1-\gamma} \geq \frac{\gamma - \gamma^{\frac{\log(2/\gamma)}{\log(1/\gamma)}}}{1-\gamma} = \frac{\gamma - \frac{\gamma}{2}}{1-\gamma} = \frac{\gamma}{2(1-\gamma)}.$$

$\square$

**Proposition 5.** *For any convex decreasing sequence $(p_t)_{t=1}^T$, if $T \geq T_\gamma + \frac{2\log(2/\gamma)}{\log(1/\gamma)}$, then there exists a valuation $v_0 \in [\frac{1}{2}, 1]$ for the buyer such that*

$$Reg(\mathcal{A}_m, v_0) \geq \max\left( \frac{1}{8}\sqrt{T - \sqrt{T}}, \sqrt{C_\gamma\left(T - \sqrt{T_\gamma T}\right)\left(\frac{1}{2} - \sqrt{\frac{C_\gamma}{T}}\right)} \right) = \Omega(\sqrt{T} + \sqrt{C_\gamma T}).$$

*Proof.* In view of Proposition 1, we only need to verify that there exists $v_0 \in [\frac{1}{2}, 1]$ such that

$$\text{Reg}(\mathcal{A}_m, v_0) \geq \sqrt{C_\gamma\left(T - \sqrt{T_\gamma T}\right)\left(\frac{1}{2} - \sqrt{\frac{C_\gamma}{T}}\right)}.$$

Let $\kappa_{\min} = \kappa^*(1)$, and $\kappa_{\max} = \kappa^*(\frac{1}{2})$. If $\kappa_{\min} > 1 + \sqrt{T_\gamma T}$, then $\text{Reg}(\mathcal{A}_m, 1) \geq 1 + \sqrt{T_\gamma T}$, from which the statement of the proposition can be derived straightforwardly. Thus, in the following we will only consider the case $\kappa_{\min} \leq 1 + \sqrt{T_\gamma T}$. Since, by definition, the inequality $\frac{1}{2} \geq p_{\kappa_{\max}}$ holds, we can write

$$\frac{1}{2} \geq p_{\kappa_{\max}} = \sum_{\kappa=\kappa_{\min}+1}^{\kappa_{\max}} (p_\kappa - p_{\kappa-1}) + p_{\kappa_{\min}} \geq \kappa_{\max}(p_{\kappa_{\min}+1} - p_{\kappa_{\min}}) + p_{\kappa_{\min}},$$

where the last inequality holds by the convexity of the sequence and the fact that $p_{\kappa_{\min}} - p_{\kappa_{\min}-1} \leq 0$. The inequality is equivalent to $p_{\kappa_{\min}} - p_{\kappa_{\min}+1} \geq \frac{p_{\kappa_{\min}} - \frac{1}{2}}{\kappa_{\max}}$. Furthermore, by Lemma 5, we have

$$\max_{v\in[\frac{1}{2},1]} \text{Reg}(\mathcal{A}_m, v) \geq \max\left(\kappa_{\max}, (T - \kappa_{\min})(p_{\kappa_{\min}} - p_{\kappa_{\min}+1})\right)$$

$$\geq \max\left(\kappa_{\max}, C_\gamma \frac{(T - \kappa_{\min})(p_{\kappa_{\min}} - \frac{1}{2})}{\kappa_{\max}}\right).$$

The right-hand side is minimized for $\kappa_{\max} = \sqrt{C_\gamma(T - \kappa_{\min})(p_{\kappa_{\min}} - \frac{1}{2})}$. Thus, there exists a valuation $v_0$ for which the following inequality holds:

$$\text{Reg}(A_m, v_0) \geq \sqrt{C_\gamma(T - \kappa_{\min})\left(p_{\kappa_{\min}} - \frac{1}{2}\right)} \geq \sqrt{C_\gamma\left(T - \sqrt{T_\gamma T}\right)\left(p_{\kappa_{\min}} - \frac{1}{2}\right)}.$$

Furthermore, we can assume that $p_{\kappa_{\min}} \geq 1 - \sqrt{\frac{C_\gamma}{T}}$ otherwise $\text{Reg}(A_m, 1) \geq (T-1)\sqrt{C_\gamma/T}$, which is easily seen to imply the desired lower bound. Thus, there exists a valuation $v_0$ such that

$$\text{Reg}(\mathcal{A}_m, v_0) \geq \sqrt{C_\gamma\left(T - \sqrt{T_\gamma T}\right)\left(\frac{1}{2} - \sqrt{\frac{C_\gamma}{T}}\right)},$$

which concludes the proof. $\square$

## 9 Simulations

Here, we present the results of more extensive simulations for $\mathsf{PFS}_r$ and the `monotone` algorithm. Again, we consider two different scenarios. Figure 3 shows the experimental results for an agnostic scenario where the value of the parameter $\gamma$ remains unknown to both algorithms and where the parameter $r$ of $\mathsf{PFS}_r$ is set to $\log(T)$. The results reported in Figure 4 correspond to the second scenario where the discounting factor $\gamma$ is known to the algorithms and where the parameter $\beta$ for the `monotone` algorithm is set to $1 - 1/\sqrt{TT_\gamma}$. The scale on the plots is logarithmic in the number of rounds and in the regret.

Figure 3: Regret curves for PFS$_r$ and monotone for different values of $v$ and $\gamma$. The value of $\gamma$ is not known to the algorithms.

Figure 4: Regret curves for PFS$_r$ and monotone for different values of $v$ and $\gamma$. The value of $\gamma$ is known to both algorithms.