[Reviews · NeurIPS 2014]

Submitted by Assigned_Reviewer_6

The paper presents an algorithm that achieves optimal regret for sellers in posted-price auctions with strategic buyers. The intuition behind the definition of Regret is not clear enough, what does a small regret mean for the seller. There should be more elaboration on the intuition. The paper is well-written with proofs and theorems clearly stated.
Summary: A good paper, proves an optimal bound which improves the previous best-known algorithm substantially.

Submitted by Assigned_Reviewer_15

The paper addresses the important issue of strategic buyers. It considers the most simple setting, of a repeated single buyer with a fixed unknown valuation. The assumption is that the buyer is strategic, but has a discounted utility.

The basic idea is that the seller can "punish" the buyer by offering each rejected price multiple times (r times). This will deter the buyer from rejecting low prices, due to the loss of utility. (However, the buyer does not become "truthful" but only optimizes its behavior.)

The paper first discusses the case of monotone pricing, where the price can never go up,
Showing a lower bound of sqrt{T} and an upper bound of sqrt{T_gamma *T}, where T_gamma = 1/(1-gamma). (I guess that the interesting range is gamma=1-1/sqrt{T} which will give a regret of T^{3/4}.)

The main contribution of the paper is an "optimal algorithm" (although I do not think that the proof shows it is exactly optimal, only that it achieves a constant from the lower bound.)

The optimal algorithm uses a pricing tree (essentially a sorted binary tree) with the idea that a rejected price is repeated r times. The buyer is given the seller strategy (the tree and r) and optimizes its utility. Essentially, in each node of the tree (price offered) the buyer compares its discounted surplus from buying and not-buying, and performs the one that gives him a higher surplus.

The regret of the seller is analyzed as a function of the discount factor of the buyer (gamma). For a discount factor of 1- 1/ log (T) the regret is logarithmic.

The paper gives a lower bound based mainly on previous works (somewhat strange to call two works which are 10 years apart, INDEPENDENT !)

There is a short synthetic empirical valuation (which is in line with the theory).
Summary: online regret minimization for single item single buyer when the buyer has a discounted utility and is strategic.

Submitted by Assigned_Reviewer_31

This paper studies an established model of repeated posted-price auctions with a single, strategic agent with an unknown value for a good. The algorithm posts a price at each time step, and the agent makes a purchasing decision based on his surplus from the current stage and the effect of his purchasing decision on future prices. In particular, the agent may strategically choose to not buy the good in the hopes of lowering the prices offered in future rounds. The authors extend an existing price-exploration algorithm for the non-strategic buyer case to this strategic case and show that their algorithm achieves near optimal strategic regret. They provide limited empirical evaluation to demonstrate superior performance over the previous work in this model.

This is a good paper and is a valuable contribution to the literature at the intersection of machine learning and algorithmic game theory. The techniques, and the associated technical exposition, are very clean and intuitive. Throughout the paper, the authors do a good job of providing the high level ideas and also do a good job of explaining the relationship between their methods and previous work. While the model studied might not be the most realistic one for the context of repeated ad auctions, the techniques and analysis in this paper are likely to helpful in the study of more realistic and complex models.

One particularly nice aspect of the approach in this paper is that the technique feels like it can be used again in similar settings. As is noted, the discount rate of the buyer is really the only aspect of this model that a designer can use to enforce good incentive properties and the algorithm in this paper seems like a general technique for leveraging the discount rate in repeated games, e.g. in a model where buyers don’t have an explicit discount rate but have some sort of ad impression budget that they must satisfy before the end of the game.

The only minor complaint I have is regards the knowledge of the discount rate. While knowledge of such parameters is standard in theoretical models, robustness to such an assumption is a desirable property (e.g. the literature on prior-free mechanism design). The authors do show a few experimental examples where their method performs well even without knowledge of the discount rate but I didn’t find that section to be particularly convincing. Perhaps more extensive empirical work could have made this point better (although that would have been difficult due to space constraints). Still this is only a minor point.

Side Notes:

- In figure 2, it is not immediately obvious that the y-axis is on a different scale. So it initially appeared that the regret with a known discount parameter was actually higher than the unknown.
Summary: A well-written paper with an elegant solution and analysis to an interesting problem at the intersection of machine learning and algorithmic game theory.
Author Feedback
Author rebuttal: We thank all reviewers for their comments.

Reviewer_15:

The reviewer is correct, we are proving "optimality" up to a log(T) factor. We will extend our discussion about this after the derivation of the lower bound. And, indeed, we should not refer to the two papers referenced as independent given the temporal gap, though our intention was only to give sufficient credit to both.

Reviewer_31:

We should first emphasize the fact that the definition of our optimal algorithm requires only the knowledge of an *upper bound* on the discounting factor \gamma. The reviewer is correct that this may not be desirable. However, the lower bound's dependence on T_\gamma shows that without an assumption on the parameter \gamma, sublinear regret is unattainable.

Reviewer_6:

Here is some more about the intuition behind the regret definition, which we could
add to our current description. The notion of strategic regret was introduced in [2] to compare the revenue obtained by a seller using a learning algorithm and the one he could have obtained with knowledge of the buyer’s valuation. The idea to compare these quantities is motivated by the following: a seller with access to the buyer’s valuation can price the object epsilon close to this value. The buyer, who remains strategic, has then no option but to accept this price in order to optimize his utility. There is no scenario where a higher revenue can be achieved by the seller, therefore it is a natural setting to compare against.